# Laboratory Measurements of the Influence of Turbulence Intensity on the Instantaneous-Fading Reciprocity of Bidirectional Atmospheric Laser Propagation Link

Yi Liu [1,2], Zhi Liu [1,2,*], Yidi Chang [1], Yang Liu [1,2] and Huilin Jiang [1,2]

1   National and Local Joint Engineering Research Center of Space Optoelectronics Technology,
    Changchun University of Science and Technology, Changchun 130022, China; liuyi@mails.cust.edu.cn (Y.L.);
    2020200108@mails.cust.edu.cn (Y.C.); 2015200024@mails.cust.edu.cn (Y.L.); hljiang@cust.edu.cn (H.J.)
2   College of Opto-Electronic Engineering, Changchun University of Science and Technology,
    Changchun 130022, China
*   Correspondence: liuzhi@cust.edu.cn

**Abstract:** The reciprocity of the atmospheric turbulence channel in the bidirectional atmospheric laser propagation link is experimentally tested. The bidirectional transceiving coaxial atmospheric laser propagation link is built by using a hot air convection-type atmospheric turbulence emulation device with adjustable turbulence intensity. The influence of different turbulence intensities on the instantaneous-fading correlation of channel is analyzed by the spot characteristics. When there is no atmospheric turbulence in the bidirectional transceiving coaxial atmospheric laser propagation link, the value of channel instantaneous fading correlation coefficient was merely 0.023, which indicates we did not find any reciprocity in the optical channel. With the increment in turbulence intensity, the channel instantaneous fading correlation coefficient presented a constant increasing trend and then tended to be stable around 0.9 in the end. At this moment, the similarity of the instantaneous change trends for these two receiving terminal optical signals, and the consistency of their probability density function, indicates that there is good reciprocity between the bidirectional atmospheric turbulence optical channels. With the increase in the optical signal scintillation factor, we can obtain the result where the correlation coefficient value decreases accordingly.

**Keywords:** atmospheric turbulence pool; bidirectional optical channel; correlation coefficient; reciprocity

## 1. Introduction

In the past few decades, atmospheric space laser communication technology has developed rapidly, which combines the characteristics of optical fiber communication and other wireless communication methods. It has the advantages of high bandwidth, flexibility, and strong anti-interference ability [1–5]. However, the random fluctuations of the refractive index of light caused by atmospheric turbulence, as well as the absorption and scattering of light signals by gas molecules in the atmosphere, etc., also need to be considered. The optical signal undergoes light intensity fluctuations, angle of arrival fluctuations, beam drift, beam expansion, wavefront distortion, and other phenomena through the atmospheric channel. This reduces the beam quality and seriously affects the performance and quality of the laser communication system [6–9]. In order to solve the above problems and suppress the effect of atmospheric turbulence, adaptive optics technology has been proposed and widely used, and this technology is also suitable for free-space bidirectional optical propagation systems [10–12]. The free-space bidirectional propagation link means that, on the same propagation path, two coaxial transmitters at different locations emit optical signals with opposite propagation directions. In this system, in order to obtain channel state information to implement an adaptive optical correction, it

is necessary to establish a dedicated channel state information feedback channel, but this has a large delay and high cost.

In a bidirectional atmospheric laser propagation link, two light waves transmitted in the opposite direction pass through the same atmospheric channel. That is, it can be considered that the two channels in the bidirectional atmospheric laser propagation link have the same fading characteristics. That is, the channels have reciprocity. In other words, when the channel is in a fixed state, the channel state of the return propagation is consistent with the channel state of the outbound propagation, and the channel state information (CSI) can be obtained at the transmitting end, so the channel is reciprocal. In this way, the adaptive optics technology can not only correct the wavefront distortion caused by atmospheric turbulence, but also reduce the additional overhead caused by establishing a dedicated channel state information feedback channel. At the same time, the reciprocity of the atmospheric turbulence channel is used for key extraction and other tasks [13]. Therefore, it is significant to measure the related characteristics of the instantaneous fading of the bidirectional atmospheric laser propagation link channel.

The theory of atmospheric channel reciprocity was first proposed by Jeffrey H. Shapiro, and the Helmholtz theory of reciprocity was extended and applied to atmospheric turbulence on a sunny day. Starting from theory, two impulse response functions (Green function) were first defined. Then, a turbulent atmospheric channel model was constructed, and later a bidirectional atmospheric laser propagation was carried out, by which the reciprocity of atmospheric turbulence channel was proved, namely point-source–point-receiver (PSPR) reciprocity. However, experimental results have shown that when the two transceivers have finite apertures, the correlation coefficient between the fluctuations of the luminous flux received at the two link ends may be significantly reduced to below 1 [14]. In a bidirectional FSO link, the received optical signals of the two transceivers composed of a single-mode fiber collimator (SMFC) can always maintain a good correlation [15,16]. At present, there is existing literature on the correlation of instantaneous fading in bidirectional atmospheric turbulence channels. Most of them first theoretically derive the analytical formula of the received optical signal or received optical power of the two terminals, and then they substitute the correlation function for calculation to obtain the correlation coefficient. Finally, numerical emulation is used to judge the influence of turbulence intensity and other factors on the correlation of instantaneous channel fading [17]. There are few reports on the correlation of instantaneous fading in bidirectional atmospheric turbulence channels under actual conditions [14,18].

In this paper, a bidirectional transceiving coaxial atmospheric laser propagation device is built, and a hot air convection type atmospheric turbulence emulation device is used as the atmospheric propagation channel for experiments. The turbulence intensity depends on the temperature difference between the upper and lower plates of the turbulence emulation pool, which realizes the channel reciprocity exploration of the bidirectional propagation and reception of coaxial atmospheric laser propagation in the adjustable atmospheric turbulence intensity channel. When the atmospheric turbulence increases from no turbulence to weak atmospheric turbulence and then to strong atmospheric turbulence, the correlation coefficient of the atmospheric channel increases from 0.023 to the maximum of 0.923, and then gradually decreases. Then, the similarity of the instantaneous change trends for these two receiving terminal optical signals, and the consistency of probability density function, indicates that there is a good reciprocity between bidirectional atmospheric turbulence optical channels. In addition, the greater the scintillation factor of the received signal, where the probability density distribution obeys the logarithmic normal distribution, the smaller the scintillation factor of the received signal, and the probability density distribution obeys the normal distribution in the bidirectional atmospheric laser propagation link, which is consistent with the unidirectional laser propagation characteristics in the atmosphere.

## 2. Theoretical Analysis

In a bidirectional transceiving coaxial laser propagation system, the laser beams emitted at the same time from the two ends are transmitted in the same atmospheric channel, but in opposite directions, and are finally received by the two receiving ends at the same time. The principle diagram of the instantaneous fading correlation measurement of the bidirectional optical signal is shown in Figure 1. We will refer to the propagation from terminal A to terminal B as the "forward propagation", and that from terminal B to terminal A as the "inverse propagation". Mark the two ends of the measurement as terminal A and terminal B, respectively. $P_A$ and $P_B$ are the received optical signals of terminals A and B, respectively. Assuming that the entire optical signal propagation process is random and stable, then the normalized cross-correlation function definition between them is [17]

$$\gamma_{AB}(\tau) = \frac{\langle P_A(t)P_B(t+\tau)\rangle - \langle P_A(t)\rangle\langle P_B(t)\rangle}{\left[\langle P_A^2(t)\rangle - \langle P_A(t)\rangle^2\right]^{1/2}\left[\langle P_B^2(t)\rangle - \langle P_B(t)\rangle^2\right]^{1/2}} \tag{1}$$

among them, $\langle\cdot\rangle$ is the mean value of the random functions $P_A(t)$ and $P_B(t)$, and $\tau$ is the instantaneous time change.

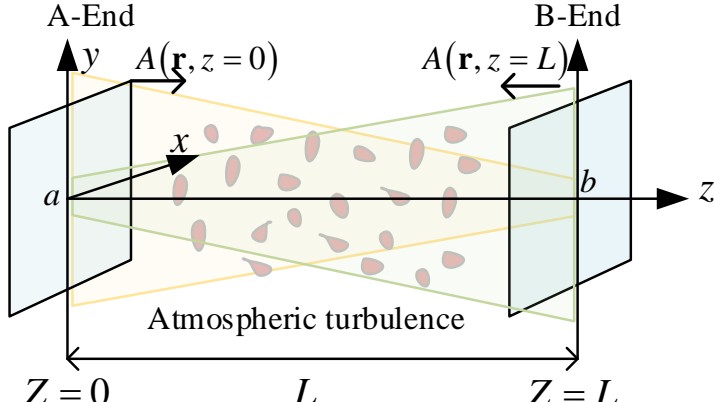

**Figure 1.** Schematic diagram of correlation measurement of bidirectional optical instantaneous fading.

The formula for calculating the light intensity scintillation factor is [19]

$$\alpha = \frac{\langle I^2\rangle - \langle I\rangle^2}{\langle I\rangle^2} \tag{2}$$

among them, $\langle\cdot\rangle$ represents the statistical average, and $I$ represents the sum of the image gray values.

Equation (1) can be used to calculate the correlation coefficient of the instantaneous fading of the optical signal received by terminal A and B, and the optical intensity flicker factor of the optical signal received by terminal A and B can be calculated by Equation (2).

## 3. Setup of the Atmospheric Turbulence Emulation Device

In order to study the correlation of bidirectional atmospheric turbulence channels under different atmospheric turbulence intensities, a hot air convective atmospheric turbulence emulation device was used as a controllable atmospheric turbulence channel [20]. This thesis adopts the hot air convection type atmospheric turbulence emulation technology, and we used the theory of thermal convection turbulence between parallel plates to design the structure [20]. The atmospheric turbulence emulation device is based on the theoretical basis of flow similarity to complete the emulation of the optical characteristics of atmospheric turbulence [21]. When the flow has similar geometric boundary conditions

and the same Reynolds number, even if the size or velocity is different or even the fluid itself is different, they will have similar dynamics [20,21].

The device mainly includes an atmospheric turbulence control system, cooling system, heating system, temperature measurement system, pool body, and window, as shown in Figure 2. The pool body is composed of high-temperature, heat-resistant, and heat-insulating plates, which are mainly used to reduce the heat exchange between the inside of the pool body and the outside world. The bottom of the pool body is a heating panel, which is evenly heated after being energized and can reach a high enough temperature to generate turbulence of different intensities. The top of the pool body is a cooling panel. The circulating flow of tap water (also available for cooling, using cooling water) keeps the cooling panel at a constant room temperature (or low temperature) to achieve different temperature differences between the upper and lower parallel plates. The temperature measurement system is composed of a temperature detector inside the tank, which can collect and record the temperature information of each part of the device in real time. The automatic control system adjusts the heating system in real time according to user preset information and temperature collection information to form a closed-loop control process.

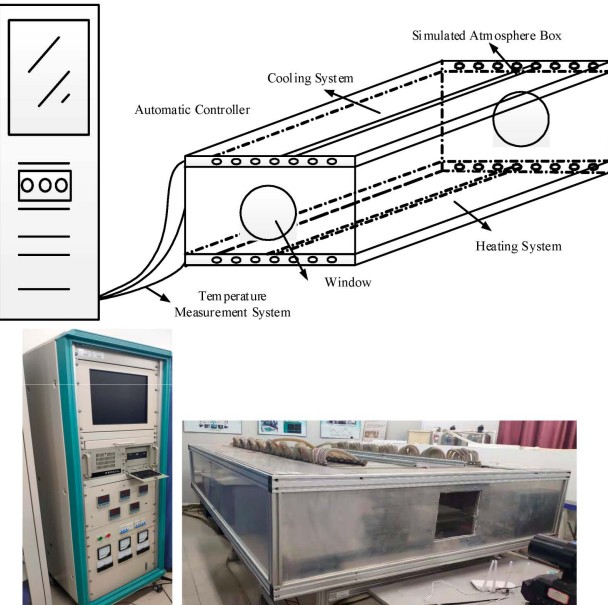

**Figure 2.** Schematic diagram of the overall structure of the hot air convection type atmospheric turbulence emulation device.

When the atmospheric turbulence emulation device is working, the lower plate is heated and the upper plate is cooled, so that the temperature difference between the two plates is gradually generated to form convection. When the temperature difference (that is, the Rayleigh number) exceeds a certain value, the flow forms turbulence, which is characterized by the physical quantity Fried constant, which is the atmospheric coherence length $r_0$. When the intensity of atmospheric turbulence increases, the atmospheric coherence length decreases. The relationship between the temperature difference of the upper and lower plates of the atmospheric turbulence emulation device and the correlation length of the atmosphere can be expressed as

$$r_0 = 48 \times (\Delta T)^{-0.81} \tag{3}$$

Therefore, the corresponding turbulence intensity can be obtained by adjusting the temperature difference between the two plates of the box. Figure 3 shows the atmospheric coherence length at a temperature ranges from 5 °C to 250 °C measured by the Hartmann method. It can be seen that the range is 0–12 cm and decreases with the increase in the

temperature difference. In this paper, the temperatures of 80, 150, and 220 °C (corresponding to $r_0$ of 1.38, 0.83, 0.57 cm, respectively) between the panels was used as the propagation channel.

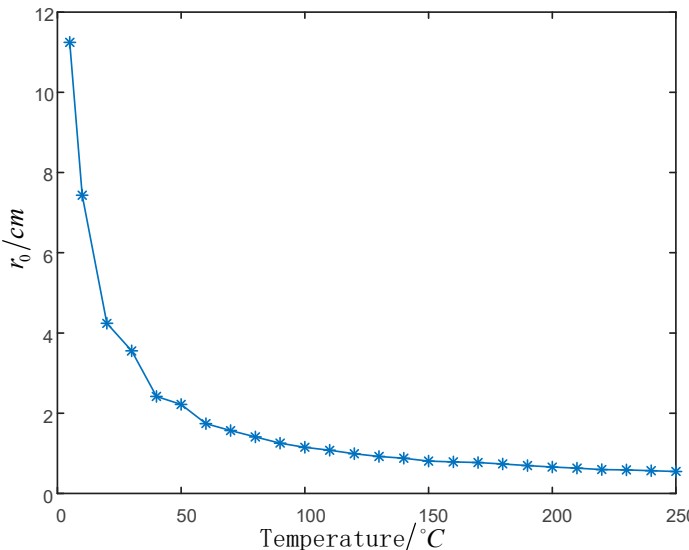

**Figure 3.** Variation of large gas coherence length with temperature difference.

## 4. Experimental Setup

The experimental system was a bidirectional transceiving coaxial atmospheric laser propagation system, as shown in Figure 4. First, the laser $l_1$ of terminal A outputs a continuous wave of 1550 nm from the port $a'$ into the single-mode fiber (SMF) circulator $c_1$, and then it outputs it through the port $b'$. After the transmissive collimating optical antenna $a_1$ expands the beam, it compresses the beam divergence and transmits it into the atmospheric turbulence channel to reach the link terminal B. The transmissive collimating optical antenna $a_2$ of terminal B receives, reduces the beam, and shapes it into parallel light. The signal enters from port $a'$ of the SMF circulator and outputs from port $b''$. The detector $d_2$ detects the light spot signal, and the acquisition card $k_2$ collects the light spot signal. Then the computer $p_2$ stores and processes the image information. At the same time, the bidirectional transceiving coaxial atmospheric laser propagation system terminal B emits the same laser beam. After the port $c''$ enters the SMF circulator $c_2$, the port outputs, through the transmissive collimating optical antenna $a_2$, the beam is expanded, the beam divergence angle is compressed, and then it is transmitted to the atmospheric turbulence emulation pool channel for propagation. Then, the transmissive collimated optical antenna $a_1$ enters the SMF circulator $c_1$ of terminal A through port $b'$, and finally outputs from port $c'$. The light spot signal is detected by the detector $d_1$, the light spot signal is collected by the acquisition card $k_1$, and the image information is stored and processed by the computer $p_1$. It should be noted that the equipment, parameters and experimental conditions of the bidirectional propagation link have always been strictly consistent during the experiment. The sampling programs of the computers $p_1$ and $p_2$ go through the synchronization server set in advance to ensure that terminals A and B sample the power of the signal synchronously, so as to ensure the reasonableness of the measurement data to the greatest extent. During the data collection process, every 10 groups of spot signals are collected, and 1 group of local spot signals are collected to ensure the reliability of the measurement data. For example, at terminal A, the signal that is emitted by laser $l_1$ and passed through SMF circulator $c_1$, and finally detected by optical power meter $d_1$, is the local spot signal of terminal A. Note that when measuring the local light spot signal, the prepared tool should be used to cut off the bidirectional atmospheric laser propagation link to remove the influence of the received light spot signal on the local optical signal. The sampling frequency of the detector was 1 kHz, the acquisition

time was 60 s, and 60,000 frames of grayscale images of the light spot were collected for each experiment.

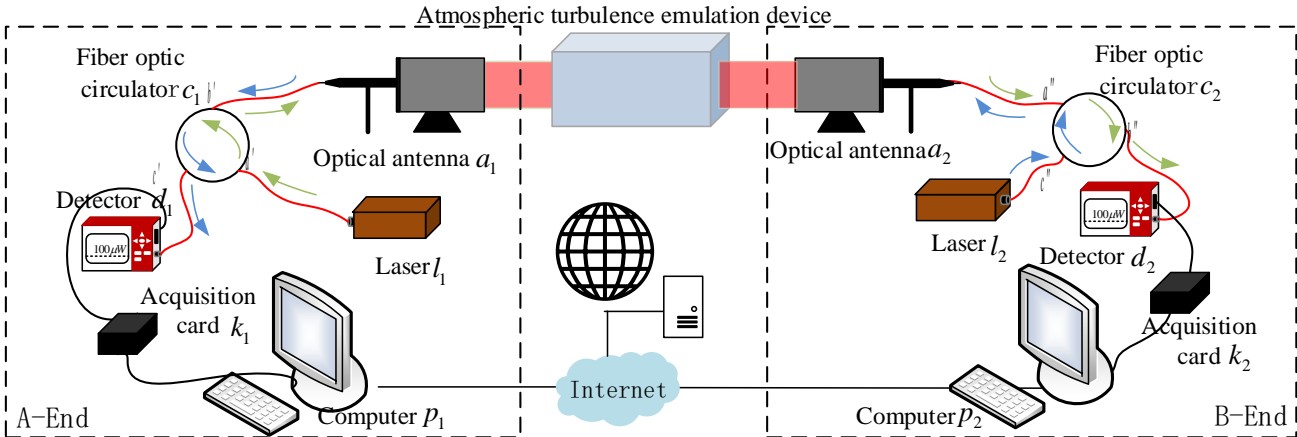

**Figure 4.** Indoor bidirectional transceiving coaxial atmospheric laser propagation system.

A high-performance continuous wave laser source was used, which works at 1550 nm and is a semiconductor laser. The isolation parameter of the SMF circulator was 40 dB. In addition, the main parameters in the experiment are shown in Table 1.

**Table 1.** Main parameters of the bidirectional coaxial atmospheric laser propagation system.

| Transmitter | |
| --- | --- |
| Aperture of optical antenna | the effective aperture: 80 mm |
| Laser | wavelength: 1550 nm<br>output power: 0–20 mW |
| SMF circulator | isolation parameter: 40 dB |
| **Atmospheric channel** | |
| Coherence length | 1–40 cm |
| Intensity frequency range | 100 Hz |
| Characteristic velocity | >0.1 m/s |
| Turbulence intensity stability | 15% |
| **Receiver** | |
| Detector | wavelength range: 185 nm–25 um<br>Power range: 100 pW–200 W |

## 5. Experimental Results and Analysis

According to the bidirectional transceiving coaxial atmospheric laser propagation link shown in Figure 4, the experimental link was built to carry out the bidirectional coaxial atmospheric laser propagation experiment. The atmospheric turbulence emulation pool was used to simulate the atmosphere as the propagation channel. The temperature gradient during the experiment was 10 °C, that is, when the temperature of the atmospheric turbulence emulation tank box increased by 10 °C, the two terminals of the bidirectional atmospheric laser propagation link simultaneously collected the power of a group of received spot signals and stored them in the computer through the acquisition card. Figure 5 shows the variation curve of the correlation coefficient of the spot signals received by terminals A and B in the propagation link with the temperature difference $\Delta T$ of the atmospheric turbulence emulation pool. We can see that when the atmospheric turbulence emulation pool was closed, that is, there was no atmospheric turbulence, the correlation coefficient of

the channel was 0.023. That is to say, the instantaneous fading of the forward and inverse propagation atmospheric turbulence channels had no correlation at this time. Since the atmospheric channel and channel state were not involved at this time, it is mainly the influence of the experimental device on the spot signal received by the two terminals of the beam spot bidirectional coaxial atmospheric laser propagation link, especially the influence of random noise on the spot signal, after opening the atmospheric turbulence emulation pool. As the internal temperature of the atmospheric turbulence pool increased with a gradient of 10, the turbulence effect gradually became more obvious. At this time, the correlation coefficients of the forward and inverse propagation propagation channels of the bidirectional coaxial atmospheric laser propagation link gradually increased. The forward and inverse propagation channel correlation coefficients increased rapidly before the temperature difference reached about $\Delta T = 50\ ^\circ$C, and the correlation coefficient of the forward and inverse propagation channels was above 0.7. As the temperature difference continued to increase, the effect became more obvious, and the correlation coefficient of the forward and inverse propagation channels gradually increased to above 0.9 and became stable. From Figure 6, one can see that the correlation coefficient of the forward and inverse propagation channels had a less obvious downward trend, and this was because as the temperature difference continued to increase, the turbulence effect in the atmospheric turbulence emulation pool gradually became stronger turbulence. At this time, the influence of atmospheric turbulence on the spot signal in the bidirectional coaxial atmospheric laser propagation link gradually increased, and the light intensity scintillation effect was more obvious. Therefore, the correlation coefficient of the forward and inverse propagation channels slightly reduced, that is, the reciprocity of the forward and inverse propagation channels slightly worsened.

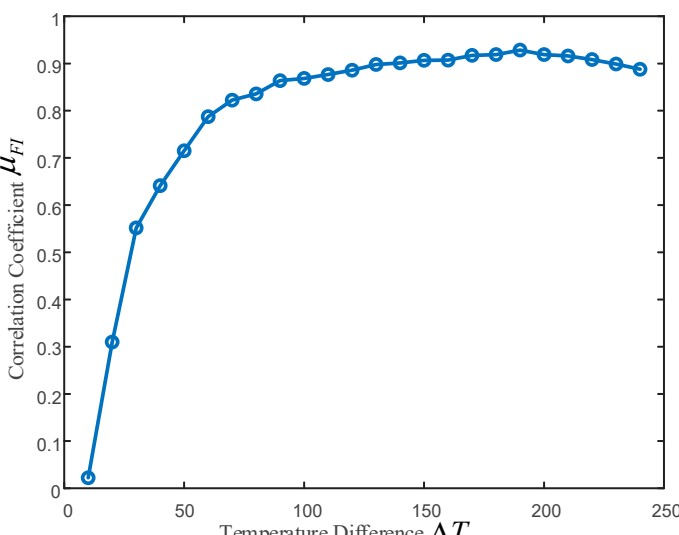

**Figure 5.** The correlation coefficient changes with the temperature difference of the atmospheric turbulence emulation pool.

Figure 6 shows the spot measurement data of the two receiving terminals of the bidirectional free-space laser propagation link under different turbulence intensities. In the experimental data under three different intensity turbulence conditions—weak, moderate, and strong—a set of representative time-domain data change curves of the spot signal of the receiving terminal are selected. The blue line in the figure is the power of the spot signal received by terminal A of the bidirectional propagation link, and the red curve is the power of the spot signal received by terminal B of the bidirectional propagation link. Figure 7a–c shows the captured time series of the light spot signal power under the conditions of weak turbulence ($T = 80\ ^\circ$C), moderate turbulence ($T = 150\ ^\circ$C), and strong turbulence ($T = 220\ ^\circ$C). It can be found that the changes to the received optical signals

of terminals A and B over time were very similar, and the correlation coefficients of the received light spot signals were calculated to be 0.9324, 0.8839, and 0.8695, respectively, indicating that the instantaneous fading characteristics of the optical signals received by the two terminals had a certain correlation.

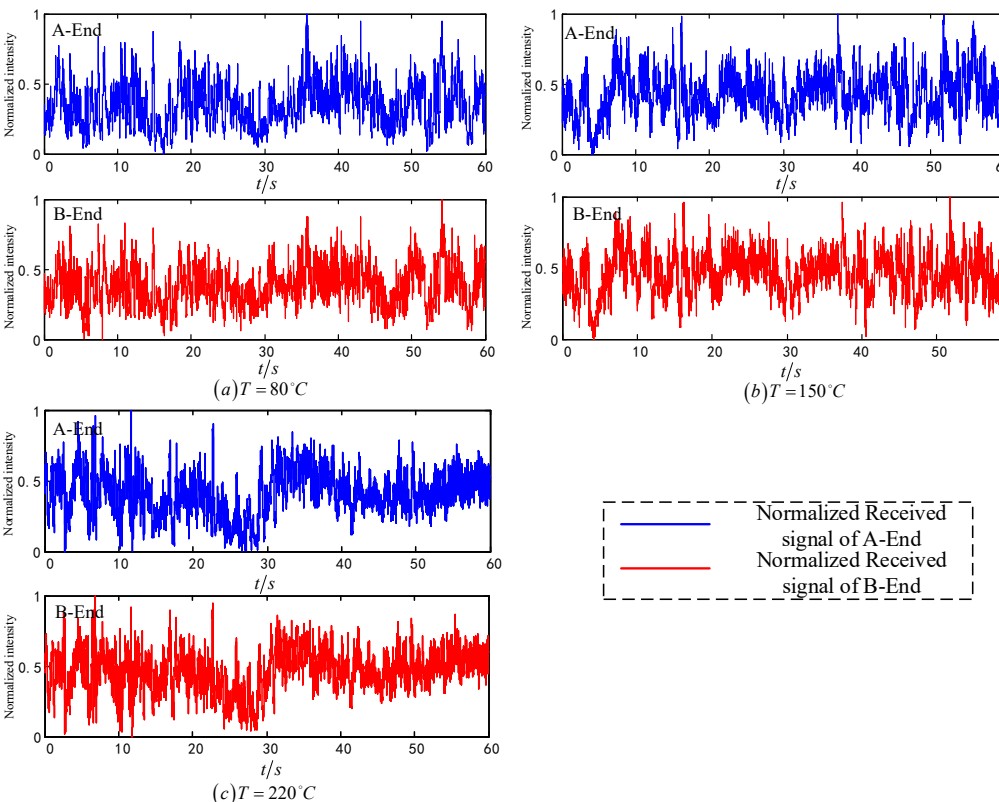

**Figure 6.** Spot measurement data of two receiving terminals of a bidirectional free-space laser propagation link under different turbulence intensities: (**a**) $T = 80\ ^\circ\text{C}$ weak turbulence, (**b**) $T = 150\ ^\circ\text{C}$ moderate turbulence, (**c**) $T = 220\ ^\circ\text{C}$ strong turbulence.

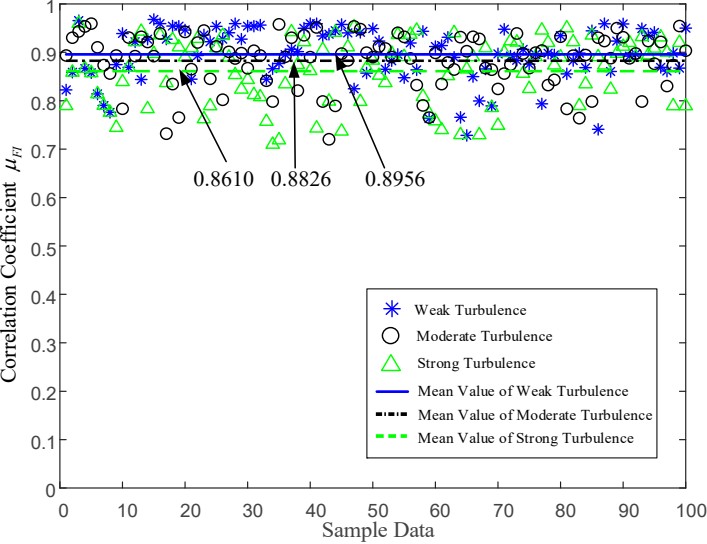

**Figure 7.** Correlation coefficient distribution and its average value.

Figure 7 shows a distribution diagram of the correlation coefficients of the optical signals received by terminals A and B and their average values under different atmospheric turbulence conditions. Using the temperature adjustable characteristics of the atmospheric

turbulence emulation pool, 100 sets of experimental data were collected under different turbulence intensities. Equation (1) is used to calculate the correlation coefficient of the received signal at both ends of the bidirectional coaxial atmospheric laser propagation link. Among them, the blue symbol is the sample data of the weak atmospheric turbulence, the black symbol is the sample data of the moderate atmospheric turbulence, and the green symbol is the sample number of the strong atmospheric turbulence. It is found that in the three sets of sample data, the correlation coefficients of the received spot signals of the two terminals were all above 0.7, and the average values were 0.8956, 0.8826, and 0.8610, respectively. Most of the correlation coefficients of the received spot signals were above or near their average line. Under different turbulence intensities, the forward and inverse propagation channels of the bidirectional coaxial atmospheric laser propagation link had good reciprocity. The intensity of atmospheric turbulence influenced the reciprocity of the forward and inverse propagation channels. The axial atmosphere laser propagation link had better reciprocity between the forward and inverse propagation channels under weak turbulence conditions. As the intensity of atmospheric turbulence increased, the channel correlation coefficient decreased slightly, that is, the reciprocity of channel deteriorated.

Figure 8 shows the spot measurement data and probability density distribution diagram of the receiving terminal of the bidirectional transceiving coaxial atmospheric laser propagation link. The received power is used to characterize the strength of the instantaneous signal at the receiver. From left to right, the normalized received signal changed with time. Terminal A normalized the received signal frequency distribution histogram and probability density distribution diagram, and terminal B normalized the received signal frequency distribution histogram and probability density distribution diagram. Among them, the green line is the measurement data of terminal A, and the red line is the measurement data of terminal B. The selected data in this group are the probability density histogram drawn by the indoor turbulence emulation device to simulate the atmospheric turbulence conditions and the samples close to the real atmospheric samples. Observing the graphs of normalized received signal changes over time in Figure 8a–c, it is found that the received signal changed over time at the A and B terminals and were very similar. That is, the instantaneous fading characteristics of the signals received by the two terminals had a certain correlation. In order to analyze the correlation between the instantaneous fading of the received signals at both ends, the measurement data represented by Figure 8a–c are calculated, and Equation (1) is used to obtain the reception at both ends of A and B. The instantaneous fading correlation coefficients $\gamma_{AB}$ of the signal are 0.8739, 0.9062, and 0.9592 in sequence. This shows that the correlation between the instantaneous fading of the received signals at both ends was very strong at this time. At the same time, Equation (2) is used to calculate the light intensity scintillation factors of the three groups of spot signals in Figure 8. The received signal scintillation factors of terminal A were 0.33895, 0.13269, and 0.01125, and the received signal scintillation factors of terminal B were 0.33659, 0.13378, and 0.01113, respectively. Combined with the frequency distribution histogram and probability density distribution diagram of the spot signal in Figure 8a–c, it can be seen that the greater the scintillation factor of the received signal, where the probability density distribution obeys the logarithmic normal distribution, the smaller the scintillation factor of the received signal, and the probability density distribution obeys the normal distribution, which is consistent with the unidirectional laser propagation characteristics in the atmosphere studied by our research group, Ni Xiaolong [22]. At the same time, it can be found that the larger the scintillation factor of the received spot signal, the stronger the turbulence intensity of the atmospheric channel, and the smaller the instantaneous fading correlation coefficient of the received spot signals of the two terminals; that is, as the turbulence intensity increases, the channel reciprocity becomes slightly worse.

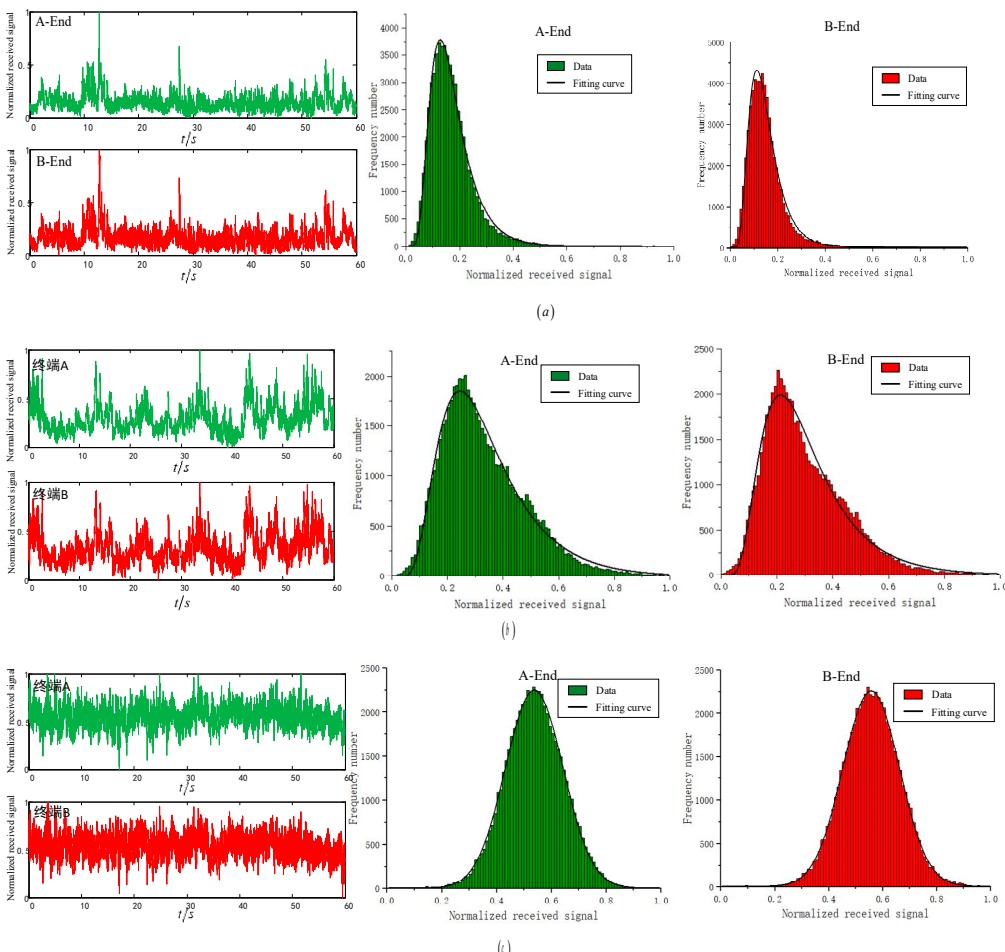

**Figure 8.** Data and probability density distribution of the receiving terminal spot measurement of the bidirectional transceiving coaxial atmospheric laser propagation link (**a**) $\gamma_{AB} = 0.8739$, $\alpha_A = 0.33895$, $\alpha_B = 0.33659$; (**b**) $\gamma_{AB} = 0.9062$, $\alpha_A = 0.13269$, $\alpha_B = 0.13378$; (**c**) $\gamma_{AB} = 0.9592$, $\alpha_A = 0.01125$, $\alpha_B = 0.01113$.

## 6. Conclusions

Aiming at the influence of atmospheric turbulence of different intensities, in this paper, a bidirectional transceiving coaxial atmospheric laser transmission link was built by using a hot air convective atmospheric turbulence emulation device as an atmospheric turbulence channel. According to the existing bidirectional atmospheric turbulence channel instantaneous fading correlation theory, a measurement method for the instantaneous fading correlation characteristics of a bidirectional atmospheric turbulence optical channel under different turbulence intensities is designed, and measurement experiments are carried out. The results show that when the atmospheric turbulence emulation device in the experiment is turned off, the instantaneous fading of the atmospheric turbulence optical channel is not relevant at this time. As the turbulence intensity increases, the correlation coefficient of the instantaneous fading of the atmospheric turbulence optical channel gradually increases, and the channel correlation coefficient increases to above 0.9 and tends to stabilize. In addition, under different atmospheric turbulence intensity conditions—that is, weak turbulence intensity, moderate turbulence intensity, and strong turbulence conditions—the received optical signals of terminals A and B have very similar changes with time. At this time, the correlation coefficients of the received light spot signals are 0.9324, 0.8839, and 0.8695, respectively, indicating that the instantaneous fading characteristics of the optical signals received by the two terminals have a certain correlation, but the stronger the turbulence, the worse the channel reciprocity. At the moment, with the increases in the normalized received optical signal scintillation factor, the correlation

coefficient of the instantaneous fading of the forward and inverse propagation atmospheric turbulence optical channels has a downward trend. The probability density distribution of the received optical signal is similar to that of the scintillation probability density function of the unidirectional atmospheric laser propagation link. It is further proved that it is feasible and meaningful to use the instantaneous fading correlation of the bidirectional optical signal to improve the adaptive optics technology in the optical communication system, and to use bidirectional atmospheric turbulence optical channel as the random source to generate the key in key extraction technology.

**Author Contributions:** Data curation, Y.L. (Yi Liu); Formal analysis, Y.L. (Yi Liu); Funding acquisition, Z.L.; Methodology, Y.L. (Yi Liu).; Project administration, Z.L.; Resources, Y.C. and Y.L. (Yang Liu); Supervision, H.J.; Writing—original draft, Y.L. (Yi Liu); Writing—review & editing, Y.L. (Yi Liu). All authors have read and agreed to the published version of the manuscript.

**Funding:** This research received no external funding.

**Institutional Review Board Statement:** Not applicable.

**Informed Consent Statement:** Not applicable.

**Data Availability Statement:** The study did not report any data.

**Conflicts of Interest:** The authors declare no conflict of interest.

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
