# Peer review of "Laboratory Measurements of the Influence of Turbulence Intensity on the Instantaneous-Fading Reciprocity of Bidirectional Atmospheric Laser Propagation Link"

_applsci, doi:10.3390/app11083499_

Round 1

Reviewer 1 Report

The authors present a thorough experiment to assess the effect of the turbulence intensity on the instantaneous fading reciprocity in bi-directional atmospheric laser links. I sincerely believe that the presented work may be of interest to the community and therefore I think it has potential for being accepted for publication providing that some issues are corrected.

  1. My main concern with this work is regarding what I believe is the most important figure of the paper. When I first looked at Figure 7, I could not observe a clear tendency since the moderate turbulence results in the lowest correlation. Reading the text, however, it seems that the labels in Figure 7 are not correct. I strongly suggest checking the labels of Figure 7.
  2. In the last paragraph of the introduction, I would be more assertive when claiming the contribution of the paper.
  3. I would not say what they are reporting is a “simulation” but an “emulation” device. Maybe it is something that depends on the area but to me “simulation” is more related to computer programming than replicating the conditions in a controlled laboratory setup.
  4. The processing is well described but there is some missing information about the optical stage of the setup. In particular, what was the operation frequency of the lasers? What kind of laser was used (semiconductor lasers, DFB, FP)? In addition, it is not clear to me whether a continuous wave source was employed or if a modulated signal was transmitted? If a modulated signal was transmitted, what kind of modulation was employed? How was the signal modulated (direct or external modulation)? Regarding the optical circulators, could the authors give some information on their characteristics? In particular the isolation parameter.
  5. I understand that the positive and negative channels are the two counter-propagating channels. I suggest using “forward” and “backward” or define the terms “positive” and “negative” in their first use.
  6. Some sentences and terms are slightly difficult to understand and could be rewritten. For instance, “time-domain variational curves” in line 238 could be written as simply “captured time series”. Another example is “Figure 7 is a distribution diagram…” in line 245, which would typically be written as “Figure 7 shows a distribution diagram” or “In Figure 7, a distribution diagram … is shown”. Another example is the sentence “Use formula (1) to calculate the correlation coefficient of the received spot signals at the two terminals of the bidirectional coaxial atmospheric laser transmission link.” in lines 250-251.
  7. There are some grammar mistakes and typos that should be corrected.
    1. Some punctuation marks, for instance, in Line 40 comma before “and” and in Line 253 the lack of comma before “respectively”.
    2. Some uses of plural. In Line 13 “different turbulence intensity”,  in Lines 152-153 “The temperature of 80ºC, 150ºC, and 220ºC … is used” among others.
  8. Finally, there are some details that must be addressed:
    1. According to the APA style, contractions must be avoided. Therefore, use “do not” instead of “don’t”
    2. In line 136, the caption of the figure includes two times “Schematic diagram”
    3. In line 165, the sentence “Enter From a port to the single-mode fiber circulator and output from e port.” must be rewritten. As a suggestion, I would identify the ports in capital letters to avoid confusion between “a” the undetermined article and the port identified as “a”. In addition, it sounds clearer to me “port A” than “A port”.
    4. Be careful with writing ºC in italics, for instance in lines 198, 199, and in other parts of the text.

Author Response

Dear Editors and Reviewers:

    Please see the attachment. Thanks.

Sincerely,

Yi Liu

Reviewer 2 Report

This is an excellent research paper and is important to the laser optical communication community.   There are several items that should be changed to make the paper a more informative and wider used paper.   The specific references for the details and parameters should be given....reference for Eq.3 and experimental parameters.   Some important optical parameters are not presented and are important for a more general use of the research results:   what are the optical parameter values such as laser beam diameter and mode, focusing lens parameters (focal length, lens diameter), size of detector pixels, specs of detector, etc.   Lastly, I suggest that the title be changed to   Laboratory Measurements of the Influence of Turbulence ...     since this is the first experimental measurement.

Author Response

(The authors gave the same response as above.)
